# Novel Roles of SPATULA in the Control of Stomata and Trichome Number, and Anthocyanin Biosynthesis

**DOI:** 10.3390/plants12030596

**Published:** 2023-01-29

**Authors:** Judith Jazmin Bernal-Gallardo, Victor M. Zuñiga-Mayo, Nayelli Marsch-Martinez, Stefan de Folter

**Affiliations:** 1Unidad de Genómica Avanzada (UGA-Langebio), Centro de Investigación y de Estudios Avanzados del Instituto Politécnico Nacional (CINVESTAV-IPN), Irapuato 36824, Mexico; 2CONACYT, Instituto de Fitosanidad, Colegio de Postgraduados, Campus Montecillo, Texcoco 56230, Mexico; 3Departamento de Biotecnología y Bioquímica, Unidad Irapuato, CINVESTAV-IPN, Irapuato 36824, Mexico

**Keywords:** SPATULA, transcription factors, leaf, trichomes, stomata, anthocyanin

## Abstract

The bHLH transcription factor SPATULA (SPT) has been identified as a regulator during different stages of Arabidopsis development, including the control of leaf size. However, the mechanism via which it performs this function has not been elucidated. To better understand the role of SPT during leaf development, we used a transcriptomic approach to identify putative target genes. We found putative SPT target genes related to leaf development, and to stomata and trichome formation. Furthermore, genes related to anthocyanin biosynthesis. In this work, we demonstrate that SPT is a negative regulator of stomata number and a positive regulator of trichome number. In addition, SPT is required for sucrose-mediated anthocyanin biosynthesis.

## 1. Introduction

Leaf formation begins with the activity of a shoot apical meristem, which, through processes of cell division, proliferation, differentiation, and expansion, contribute to the final size and shape of the leaf [1,2,3]. In addition, specialized cells called stomata and trichomes develop on the leaf. The formation of stomata is initiated from meristemoid mother cells (MMCs), these cells divide asymmetrically producing a meristemoid (M) and a larger sister cell named a stomatal lineage ground cell (SLGC). Subsequently, meristemoid cells may undergo asymmetric amplifying divisions before differentiating into a guard mother cell (GMC). SLGCs may differentiate into a pavement cell or reacquire a MMC fate and initiate asymmetric spacer divisions that produce “satellite meristemoids”. Stomatal complexes are formed after at least one unequal division of a mother cell, and then by a single equal division of a MMC [4,5,6]. Stomatal development requires sequential activity of the basic helix–loop–helix (bHLH) transcription factors SPEECHLESS (SPCH), FAMA, and MUTE, which form heterodimers with SCREAM1 (SCRM1) and SCRM2. Three peptide-receptor kinase signaling pathways enforce correct stomatal patterning, where the peptides EPIDERMAL PATTERNING FACTOR1 (EPF1) and EPF2 act as negative regulators of the MUTE–SCRM and SPCH–SCRM modules, respectively. On the contrary, the peptide STOMAGEN (or EPFL9) acts as a positive regulator of the SPCH–SCRM module [6,7,8,9,10,11].

Trichome formation begins when protodermal cells stop cell division and proceed to endoreplication; after two or three rounds of endoreplications, the trichome begins to expand out of the leaf surface and initiates two branching events, before the formation of a mature trichome on the leaf [12]. Both stomata and trichomes contribute to proper leaf function, allowing the leaf to properly perform photosynthesis and gas exchange, in addition to outward protection and sensing [6,7,8,9,10,12,13].

On the other hand, the production of compounds such as anthocyanins occurs in the plant. Anthocyanins are pigments that belong to the group of flavonoids and accumulate in different parts of the plant, including leaves, stems, flowers, and fruits [14]. These compounds are involved in the attraction of pollinators and seed dispersers. They also play a protective role against different biotic and abiotic stresses [15]. The anthocyanin biosynthesis pathway is regulated by different factors such as light and sucrose. Light and high concentrations of sucrose both induce the accumulation of anthocyanins [16,17].

It is known that trichome formation and anthocyanin biosynthesis are processes regulated by transcription factors belonging to three different families, through the formation of the protein complexes called MBW (MYB-bHLH-WD). In Arabidopsis, the WD40 repeat protein TRANSPARENT TESTA GLABRA1 (TTG1) and the bHLH transcription factors GLABRA3 (GL3) and ENHANCER OF GL3 (EGL3) are present in both MBW complexes, while the MBW complex involved in trichome initiation also includes the bHLH transcription factor MYC1 and the R2R3 MYB transcription factors GLABRA1 (GL1) and MYB23 [13,18]. Instead, the MBW complex involved in anthocyanin biosynthesis includes the R2R3 MYB transcription factors PRODUCTION OF ANTHOCYANIN PIGMENT 1 (PAP1), PAP2, MYB113, MYB114, and TRANSPARENT TESTA2 (TT2) and the bHLH transcription factor TRANSPARENT TESTA 8 (TT8). Moreover, anthocyanin biosynthesis requires several genes that code for enzymes, which are grouped into early biosynthetic genes (EBG) and late biosynthetic genes (LBG). The LBG are regulated by the MBW complex [18,19,20].

One of the transcription factors involved in leaf size regulation is the transcription factor SPATULA (SPT) [21]. SPT belongs to the basic helix–loop–helix family of transcription factors [22]. This family of transcription factors has been identified to be involved in different developmental processes such as trichome formation, stomata differentiation and proliferation, anthocyanin metabolism, light signaling, and the development of different aerial tissues in Arabidopsis [23]. In particular, SPT has been attributed to roles in flower and gynoecium development, germination, root elongation, and leaf size, as well as its relationship with PHYTOCHROME-INTERACTING FACTORS (PIFs) and far-red light perception for gynoecium development [21,22,24,25,26,27,28,29,30,31,32,33,34,35]. However, its broad expression pattern in different tissues and developmental stages gives us insight into other possible functions in Arabidopsis development.

SPT acts as a negative regulator of leaf formation, as the *spt-2* mutant has bigger leaves than wildtype (WT); however, the mechanism via which it performs this function in the leaf has not been fully elucidated. Furthermore, the expression patterns in stomata, leaf xylem, hypocotyls, and leaf lamina suggest possible functions of SPT that are not yet known [21,30]. Therefore, the objective of this work was to identify putative target genes of SPT and to determine in more depth its role in leaf development in Arabidopsis. We found that SPT participates in the control of stomata and trichome number, and it is a positive regulator in the control of anthocyanin biosynthesis.

## 2. Results

In order to gain a deeper understanding of the role of the bHLH transcription factor SPATULA (SPT) in leaf development, we identified potential SPT target genes through a transcriptomic analysis using an inducible SPT line. The inducible *35S::SPT:GR* line [27] was used. After 4 h of 10 µM dexamethasone (DEX) treatment, total RNA was isolated from 3 week old rosette leaves. RNA-seq analysis was performed; using edgeR with an FDR cutoff ≤0.05, we detected 1736 upregulated and 1960 downregulated genes (Appendix A).

Using GO analysis of the differential expressed genes (DEGs), we found enriched categories related to leaf tissues, leaf vascular tissue, and leaf structure, which is consistent with previously reported functions of SPT in leaf development. In addition, the category related to guard cells was also enriched, suggesting that SPT could be involved in stomata formation. On the other hand, categories related to flower, carpel, root, and embryo tissues were observed; tissues where SPT is known to be important for their development (Figure 1A, Appendix A).

Furthermore, we observed some putative target genes that code for transcription factors involved in leaf development, as well as in light, cytokinin, and auxin signaling, functions in which SPT has been reported to be involved. Interestingly, transcription factors related to trichome development and flavonoid functions were also found. Among them, transcription factors such as GLABRA 2 (GL2) and TRANSPARENT TESTA 8 (TT8) related to trichome formation, as well as PRODUCTION OF ANTHOCYANIN PIGMENT 1 (PAP1) involved in anthocyanin production, suggest possible novel functions for SPT in leaf development (Figure 1B).

Furthermore, by analyzing other SPT target genes, we also identified some genes related to the formation of or expressed in stomata, trichomes, and flavonoid biosynthesis; thus, it was interesting to analyze in more detail these possible novel SPT functions in leaves (Figure 2).

On the other hand, upon obtaining the promoter region of the putative SPT target genes (500 bp upstream of the ATG), in silico analysis was performed to identify the statistically significant enriched DNA-binding motifs in the promoter regions of these sequences (Figure 1C). The analysis showed the G-box DNA-binding motif characteristic for the bHLH transcription factor family to which SPT belongs, in the promoters of PHYTOCHROME-INTERACTING FACTORS (PIFs) and MYCs, suggesting that some target genes identified could be direct SPT target genes. In addition, binding sites for transcription factors of the MYB, YABBY, TCP, auxin response factor (ARF), and Arabidopsis response regulator (ARR) families were identified, suggesting that members of these families together with SPT might be regulating these target genes (Figure 1C, Appendix A).

### 2.1. SPT Contributes to the Control of Stomata and Trichome Number

To establish a possible role of SPT in stomata and trichome development, these structures were counted in different backgrounds. For stomata counting, a statistically significant increase in the number of stomata was observed on both sides of *spt* mutant leaves (abaxial: 16.88 ± 3.08 stomata per view field, adaxial: 13.46 ± 2.62 stomata) compared to wildtype (WT) (abaxial: 13.47 ± 2.06 stomata, adaxial: 10.68 ± 1.95 stomata). In contrast, no statistically significant difference was observed in the *35S::SPT* line (abaxial: 14.88 ± 1.96 stomata, abaxial: 10.81 ± 1.47 stomata) compared to WT. This suggests that SPT negatively regulates stomata number; however, on the basis of the results from the overexpressing line, SPT is not sufficient to regulate stomata number on its own (Figure 3A–F,J–K). Counting was performed in a similar area.

On the other hand, the number of trichomes on the adaxial side of *spt* mutant leaves (0.36 ± 0.06 trichomes per mm^2^) was statistically significantly lower compared to WT (0.52 ± 0.17 trichomes per mm^2^), while, in the SPT overexpression line (0.48 ± 0.07 trichomes per mm^2^), there was no significant change with respect to WT, suggesting that SPT positively regulates leaf trichome number (Figure 3G–I,L). However, on the basis of the results from the overexpressing line, SPT is not sufficient to regulate the stomata number on its own and probably needs an interactor. Counting was performed in a similar area.

### 2.2. SPT Is Required for the Control of Sucrose-Mediated Anthocyanin Induction

Anthocyanin production in Arabidopsis is regulated by different factors such as sucrose, which stimulates anthocyanin accumulation in seedlings. Because of the previously reported relationship between SPT and anthocyanin production [37] and the genes related to the production of anthocyanin (Figure 2), we analyzed the possible role of SPT in the production of anthocyanins stimulated by sucrose (Figure 4). The assay showed that the *spt-2* mutant accumulated fewer anthocyanins compared to WT when grown on MS medium supplemented with 5% sucrose, mainly in the upper part of the Arabidopsis hypocotyl (Figure 4A–F). We did not observe this reduced accumulation for the *spt-12* mutant allele. We measured the presence of anthocyanin with the mean value of red–green–blue (RGB) in the images, and we observed the different patterns of these colors in the samples (Figure 4G,H). Moreover, we obtained the mean intensity value showing that, in *spt-2* at 1% sucrose as a control, no intensity differences were observed with respect to WT, but the *SPT* overexpression (*35S::SPT*) showed higher intensity than both WT and *spt-2*. However, when the seedlings were grown in excess of sucrose at 5%, we observed that both WT and *35S::SPT* seedlings have a higher color intensity than *spt-2* (Figure 4I). Interestingly, in both sucrose conditions, the *spt-2* mutant seems to be unaffected by the increase in sucrose. In summary, these results suggest that SPT is required for the control of sucrose-mediated anthocyanin induction, which correlates with the transcriptomic data.

## 3. Discussion

The reported functions of the bHLH transcription factor SPT have mainly been related to the development of the gynoecium, flower, and seeds [24,25,28,29,30,31,32,33,34,35]. However, it is known that SPT is important for leaf size control, and its expression is observed in stomata [21]. In this work, we identified SPT putative target genes in the context of the leaf using RNA-seq and part of these identified genes are related to stomata and trichome number, as well as to flavonoid production, especially anthocyanins, suggesting that SPT participates in these functions during leaf development.

Previous studies have indicated that SPT requires association with other transcription factors to carry out its functions in the gynoecium [28,29]. In addition, SPT is capable of forming dimers with transcription factors belonging to the bHLH family such as INDEHISCENT (IND), ALCATRAZ (ALC), HECATE 1, 2, 3 (HEC 1, 2, 3), and BRASSINOSTEROID-ENHANCED EXPRESSION 1 (BEE1), and with members of other families such as ARR14, SHATTERPROOF2 (SHP2), and YAB3 [28,31,38,39]. Therefore, we performed an in silico analysis of the promoter regions of the putative SPT target genes to identify transcription factor binding sites (TFBSs). The bHLH family of transcription factors is characterized by a CACGTG DNA BS, called G-box [40,41,42,43]. It has been reported that different members of the bHLH family are capable of binding to the G-box, such as MYC2, MYC3, MYC4, PIF3, PIF4, and PIF5 [36]. Moreover, SPT is able to bind to promoter regions through the G-box of *PINOID*, *WAG2*, and *PHYTOCHROME INTERACTING FACTOR 3-LIKE PROTEINS* (PILs) genes [28,36,43]. Accordingly, during the in silico analysis, an enrichment of G-boxes was observed in the promoter regions of some putative SPT target genes, suggesting that these genes could be direct targets.

We found TFBSs for different families of transcription factors related to leaf growth such as the YABBY family genes that have been associated with adaxial and abaxial leaf formation and expansion [44], not only in Arabidopsis but also in other species [45,46,47]. Another important TFBS family identified is the TEOSINTE BRANCHED1/CYCLOIDEA/PROLIFERATING CELL FACTOR (TCP) transcription factor family that has been characterized to participate in leaf size control and act in cyclin activation for leaf endoreplication [48,49,50]. In addition, we identified TFBSs for auxin (ARF3/ETTIN) and cytokinin (ARR14 and ARR11) signaling response; both hormones are important for leaf development, expansion, proliferation, differentiation, and leaf size establishment [51,52,53,54,55,56,57,58,59,60]. Overall, this suggests that SPT is capable of regulating its target genes potentially through the formation of protein complexes with members of its own family, as well as with members of other transcription factor families.

In this work, we focused on evaluating novel functions of SPT. Transcriptomic analysis suggests the possible involvement of SPT in the formation of stomata and trichomes, and the biosynthesis of anthocyanins. Among the SPT target genes that gave us an indication of this is the repression of STOMAGEN expression by SPT induction. STOMAGEN is a peptide that acts positively in the regulation of stomata formation [61,62]. STOMAGEN has been described as a direct target gene positively regulated by the transcription factor ELONGATED HYPOCOTYL 5 (HY5) [11]. In addition, STOMAGEN is directly repressed by the transcription factor MONOPTEROS (MP), and its repression depends on the presence or absence of auxins [58]. In order to test the involvement of SPT in stomata formation, we analyzed the number of stomata present in Arabidopsis leaf areas and found that the *spt* mutant produced a higher number of stomata. This suggests that SPT is a negative regulator of stomata number, and this regulation could be through the repression of STOMAGEN. Importantly, although we found more genes related to stomata development, which are related to stomatal movement, it will be interesting to investigate in future studies whether this function is related to SPT.

On the other hand, we found in the *spt* mutant a lower number of trichomes compared with WT, which could be related to the target genes of SPT, since one of the transcription factors is essential for trichome formation. GL2 is known to be involved not only in trichome development, but also in other developmental processes in different organs of Arabidopsis [18]. These different functions are due to the diversity of transcription factor complexes that activate or repress this transcription factor involved in different processes. In relation to trichome formation, GLABRA 1 (GL1) and GLABRA 3 (GL3) can bind to the GL2 promoter, which could be direct regulators of GL2 [18]. Therefore, GL2, which is in the regulatory pathway of SPT, might not be a direct target gene of SPT.

Anthocyanin biosynthesis requires several genes that code for enzymes, which are grouped into early biosynthetic genes (EBG) and late biosynthetic genes (LBG). The LBGs are regulated by transcription factors belonging to three different families through the formation of complexes called MBW (MYB-bHLH-WD) [18,19,20]. In this work, we observed that the induction of SPT positively affects the expression of *PAP1* and *TT8*; both protein coding genes are part of the MBW complex necessary for anthocyanin biosynthesis. In addition, the genes *DIHYDROFAVANOL REDUCTASE* (*DFR*), *UDP-GLUCOSYL TRANSFERASE 79B1* (*UF3GT*) [63], *CYTOCHROME P450 75B1* (*F3’H/TT7*) [64], and *UDP-GLUCOSYL TRANSFERASE 78D2* (*UGT78D2*) [65] that code for different enzymes involved in anthocyanin biosynthesis were also induced by SPT. Furthermore, the anthocyanin biosynthesis pathway is regulated by different factors such as sucrose and light. A high concentration of sucrose induces the accumulation of anthocyanin [16], while light also induces the accumulation of this pigment [17]. Both inducing factors act through the regulation of PAP1, such that, in the *pap1*/*myb75* mutant, the anthocyanin accumulation induced by sucrose and light decreases drastically compared with WT [16,66]. We observed that sucrose-induced anthocyanin accumulation in the *spt-2* mutant was reduced compared with the WT, suggesting that this process is partially dependent on SPT; according to the transcriptomic data, this regulation could be through the induction of the *PAP1*, *TT8*, *DFR*, *UF3GT*, *TT7*, and/or *UGT78D2* genes.

On the other hand, it has been reported that PIF3 positively regulates [67], while PIF4 and PIF5 negatively regulate light-induced anthocyanin accumulation, through direct binding to the promoter regions of genes involved in the anthocyanin biosynthesis pathway [68]. Moreover, SPT, PIF4, and PIF5 share diverse target genes, which are regulated through binding to the same promoter regions [27]. These data suggest that SPT and PIF regulate the anthocyanin biosynthesis pathway through the same genes, with SPT and PIF3 acting positively, but PIF4 and PIF5 acting negatively.

In addition, it is noteworthy that transcription factors such as GL2 have also been associated with the regulation of flavonoid production, specifically anthocyanins [18,69]; interestingly, SPT has also been associated with being involved in flavonoid production through in silico predictions of anthocyanin production-related networks [37]. SPT appears in one of these networks along with the transcription factors PAP1 and TT8, which are transcription factors positively involved in anthocyanin production. Moreover, they are genes positively regulated by SPT according to our transcriptome data; furthermore, we observed in the *spt-2* mutant a decrease in anthocyanin production upon increasing sucrose in Arabidopsis seedlings.

## 4. Materials and Methods

### 4.1. Growth Conditions

The *35S::SPT:GR* line from Charlie Scutt [27] and Col-0 were grown under greenhouse conditions for RNA-seq material. The lines Col-0, Ler, *spt-2* (CS275), *spt-12*, and *35S::SPT* from Karen Halliday [34] were grown under long-day conditions (16 h/8 h light/dark) at 22 °C in a growth chamber for the stomata, trichome, and anthocyanin experiments.

### 4.2. RNA Extraction

Three weeks after germination, the SPT transcription factor (*35S::SPT:GR*) was induced with 10 µM dexamethasone solution with 0.015% Silwet in distilled water. After 4 h of the induction, the first two rosette leaves per plant were collected in triplicate and frozen in liquid nitrogen. Subsequently, RNA was extracted with the OMEGA-BIOTEK E.Z.N.A Plant RNA kit and resuspended in DEPC water.

### 4.3. RNA-Seq and Data Analyses

Library preparations were performed using the TruSeq 2.0 protocol. Sequencing was paired-end with 100 bp, with an Illumina Hiseq 4000 sequencer. The libraries were prepared and sequenced at Macrogen Inc. in South Korea. On average, 14 M reads were obtained per library. Once the sequencing fastq files were obtained, the quality of the reads was verified with the FastQC program (https://www.bioinformatics.babraham.ac.uk) according to the parameter ≥QC30 of each transcript per sample. Alignment and quantification of transcripts was performed with the kallisto program [70] by aligning the reads against *Arabidopsis thaliana* transcripts found in the Araport11 archive (https://www.araport.org). On the basis of PCA analysis, subsequent analysis was performed on duplicate samples. Differential expressed genes (DEGs) were identified using the edgeR package with the contrast being *35::SPT:GR* vs. Col-0 with an FDR cutoff ≤0.05. Furthermore, we performed an in silico analysis in SHINY GO (http://bioinformatics.sdstate.edu/go/ accessed on 1 January 2022) using enrichment of the Plant Ontology Pathway database with an FDR of 0.05. For promoter analysis, we used MEME suite 5.4.1 (https://meme-suite.org) with 500 bp promoter regions upstream of the ATG of each putative SPT target gene obtained from the Arabidopsis archive (https://www.arabidopsis.org), followed by the Simple Enrichment Promoter Analysis of motifs (SEA) [71] (https://meme-suite.org) using PMB motifs of Francisco-Zorrilla, et al. (2014) [36] as a motif database, with an E-value ≤10 and the adjusted *p*-value.

### 4.4. Aerial Measurement and Trichome Count

At 20 days after germination (DAG) in soil, images were taken of the first and second youngest rosette leaves using a Leica EZ4D stereoscope. ImageJ software was used for image analysis. Trichome counts per area were evaluated using a one-way ANOVA Tukey honestly significant difference (HDS) statistical analysis with a *p*-value of 0.05 in R. The number of samples was *n* = 10.

### 4.5. Microscopic Analysis of Stomata

At 6 DAG in ½ MS growth medium, the cotyledons were collected and dehydrated with 75%, 85%, 95%, and 100% ethanol dilutions for 2 h incubation each, and then left in Hoyer’s solution for 2 days. Subsequently, they were observed using a Nomarski Leica DM4000 microscope with DIC function, analyzing the abaxial and adaxial part of the cotyledons; counting was conducted in a viewing field obtained using a 10× ocular and with a 40× objective (area: 28.13 μm × 17.58 μm). Data were analyzed using a one-way ANOVA and Tukey HSD test with a *p*-value of 0.05 in R. The number of samples was *n* = 20.

### 4.6. Sucrose-Induced Anthocyanin Accumulation Assay

Seeds were surface-sterilized and plated on half-strength MS medium, pH 5.8. The medium was solidified with 1% plant agar and supplemented with 1% (control) or 5% sucrose (anthocyanin induction). Seeds were cold-stratified on plates for 3 days at 4 °C in the dark. Next, plates were incubated in a growth chamber with a photoperiod of 16 h light/8 h dark at 22 °C. The images were taken 3 days after germination using a Leica EZ4 D stereomicroscope (Leica, Wetzlar, Germany). We performed an analysis of color measurement of red, green, and blue (RGB), as well as the intensity mean value with image J in Arabidopsis seedlings of WT, *spt-2* mutant, and *35S::SPT* in the hypocotyl zone in an area of 30 × 5 pixels (in triplicate). Data were analyzed using a one-way ANOVA and Tukey HSD test with a *p*-value of 0.05 in R.

## 5. Conclusions

In this study, we found that SPATULA (SPT) is a bHLH transcription factor that, in addition to regulating leaf size, is involved in the formation of specialized cells in the leaf such as trichomes and stomata. It is a positive regulator of trichome number and a negative regulator of stomata number, speaking to its dual function of inducer and repressor as a transcription factor. In addition, in both cases, the results suggest that SPT needs a protein interactor to perform these functions. Furthermore, SPT is a positive regulator of anthocyanin production.

Analyzing the putative target genes of SPT, we found transcription factors related to leaf formation and other developmental events that have been described for SPT such as light perception, as well as auxin and cytokinin signaling, which have already been described, also contributing in some way to leaf formation. Furthermore, by analyzing the promoters of these putative SPT target genes, we found DNA-binding motifs related to families of transcription factors that have been related to leaf formation, giving us an idea of the regulatory pathway in which SPT is participating to contribute to leaf development in Arabidopsis.

The data generated in this work contribute to the discovery of new functions of SPT and evidence what has already been described for this transcription factor. Having a long list of target genes related to different developmental events, it will be interesting to continue discovering how SPT is important in Arabidopsis leaf development and possibly other organs.

## Figures and Tables

**Figure 1 plants-12-00596-f001:**
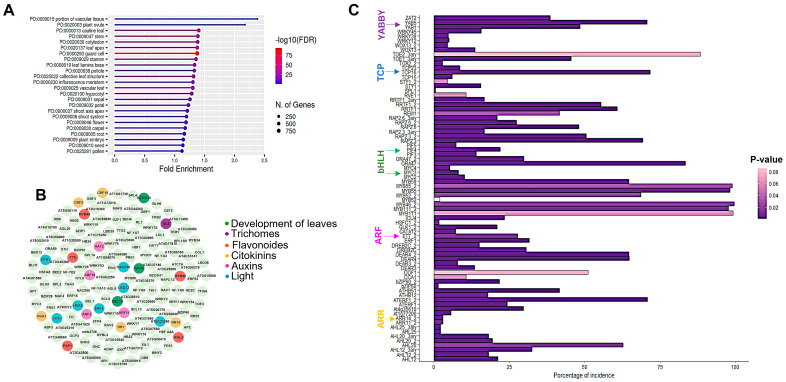
Putative target genes of SPT in leaves are related to the development of leaves, trichomes, anthocyanin, and other processes. (**A**) Enrichment analysis based on GO plant category, showing categories related to leaves, guard cells, and aerial parts of the plant. (**B**) Putative target genes of SPT that code for transcription factors related to leaf development and other developmental processes described for SPT involvement. (**C**) Result of enrichment promoter analysis of putative target genes of SPT that shows domains related to transcription factor families related to various processes, e.g., leaf development based on the Francisco-Zorrilla 2014 database [36].

**Figure 2 plants-12-00596-f002:**
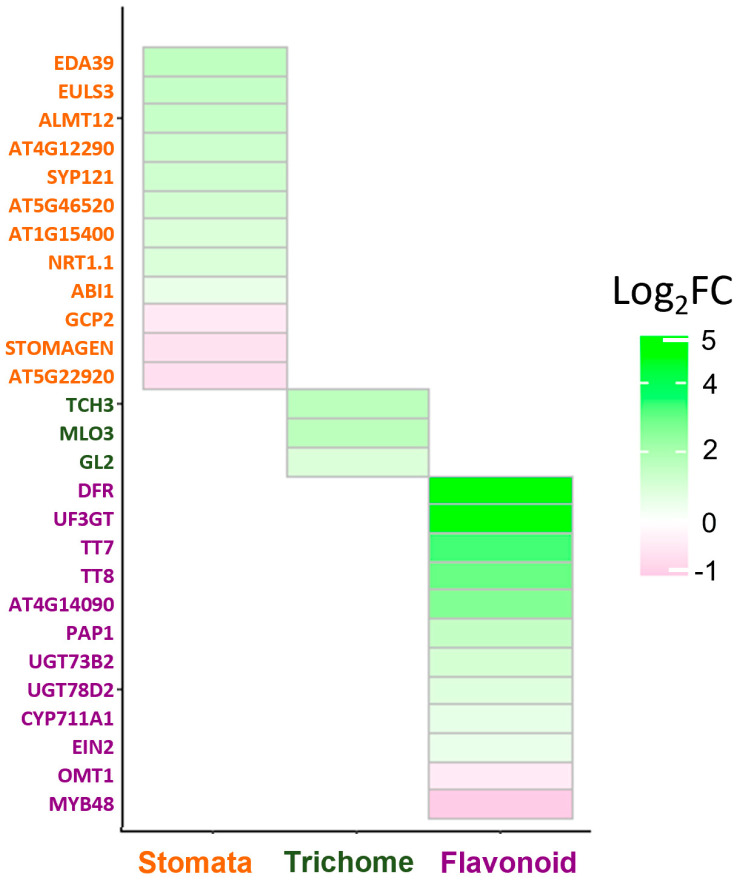
Heatmap of putative target genes of SPT related to stomata, trichome, and flavonoid biosynthesis. The scale bar represents the log_2_ fold change of putative SPT target gene expression induction in leaves compared with wildtype.

**Figure 3 plants-12-00596-f003:**
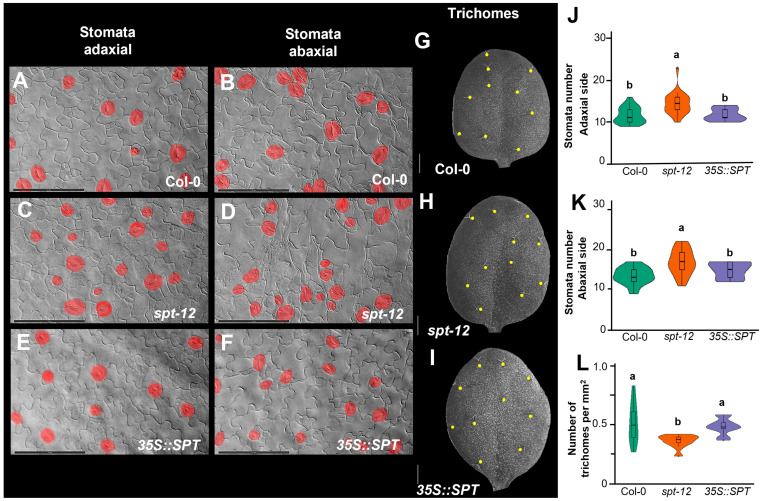
SPT is implicated in the development of stomata in cotyledons and trichomes on leaves. (**A**) Adaxial part of a cotyledon of wildtype (WT). (**B**) Abaxial part of a cotyledon of WT. (**C**) Adaxial part of a cotyledon of *spt-12.* (**D**) Abaxial part of a cotyledon of *spt-12*. (**E**) Adaxial part of a cotyledon of *35S::SPT*. (**F**) Abaxial part of a cotyledon of *35S::SPT*. (**G**–**I**) Adaxial part of leaves of WT, *spt-12*, and *35S::SPT*, respectively. (**J**,**K**) Quantification of stomatal number of the adaxial and abaxial part of cotyledons in WT, *35S::SPT*, and *spt-12.* (**L**) Quantification of the trichome number of the adaxial part of leaves in WT, *35S::SPT*, and *spt-12.* The yellow points in (**G**–**I**) represent the location of leaf trichomes. Statistical differences detected by one-way ANOVA and Tukey HSD at *p* < 0.05 are represented by different lowercase letters above the boxplots. Number of analyzed samples: *n* = 20 samples for cotyledons, *n* = 10 samples for trichome quantification. Scale bars: (**A**–**F**) 10 µm, (**G**–**I**) 1 mm.

**Figure 4 plants-12-00596-f004:**
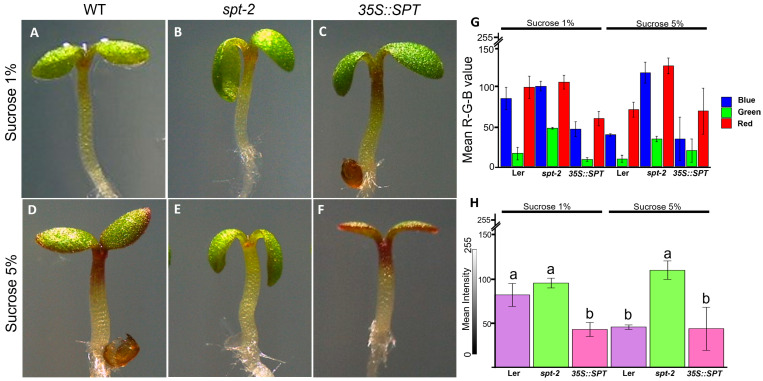
SPT is required for sucrose-mediated anthocyanin induction in seedlings. (**A**–**C**) Seedlings of WT, *spt-2*, and *35S::SPT* exposed to 1% sucrose. (**D**–**F**) Seedling of WT, *spt-2*, and *35S::SPT* exposed to an excess of sucrose at 5%. (**G**) Mean value of the intensities of red, green, or blue (R, G, or B, respectively) colors in WT, *spt-2*, and *35S::SPT* in an area of 30 × 5 pixels. (**H**) Mean value of the intensity in WT, *spt-2*, and *35S::SPT* in an area of 30 × 5 pixels. Statistical differences detected by one-way ANOVA and Tukey HSD at *p* < 0.05 are represented by different lowercase letters above the bar and histogram plots. The analyses were performed in triplicate.

## Data Availability

The RNA-seq data presented in this study are openly available in the European Nucleotide Archive (ENA), reference number PRJEB58994.

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
