# Peer review of "Novel Roles of SPATULA in the Control of Stomata and Trichome Number, and Anthocyanin Biosynthesis"

_plants, 2023, doi:10.3390/plants12030596_

Round 1

Reviewer 1 Report

Bernal-Gellado et al. investigate in this manuscript the role of the SPATULA transcription factor in plant development and identify new functions related to the control of stomata and trichome number as well as anthocyanin biosynthesis. They started their work by identifying genes responding to the activation of a SPT-GR fusion protein and then characterize process that appear to be affected by SPT. This is a perfectly valid strategy and one that in this case has led to several new discoveries that improve the understanding of this essential transcriptional regulator.

I have only minor suggestions for improvements:

Lines 127 – 135: “DNA binding domain” – do you mean “DNA binding motifs”? It would probably important to say that G-boxes are not only bound by bHLH transcription factors but also by bZIPs, etc. From the description it was not clear how putative binding sites were identified. Were they significantly enriched in the dataset or just observed? I recommend clarifying and/or better describing this analysis.

Line 160: “SPT is involved in anthocyanin biosynthesis” – this sounds as if SPT played a direct role in anthocyanin biosythesis. It would be probably better to say “SPT is involved in the control of anthocyanin biosynthesis”, or something along those lines.

Figure 3: trichome and stomata counts. From panels G-I it appears that Col leaves may be smaller than those of the spl mutant tested. If true, then an increase of stomata number would not be unexpected. In fact, in that case one would conclude that SPT has an effect on leaf size rather than stomata number. Perhaps a calculation of number of stomata per surface area does the trick.

Spt alleles: is there any reason why different spt alleles have been used for this work? Are the described phenotypes also observed in other spt alleles, i.e. are they allele specific?

Writing: I feel that the text would benefit from another round of editing – the clarity of the text could be improved in some sections. 

Author Response

Bernal-Gellado et al. investigate in this manuscript the role of the SPATULA transcription factor in plant development and identify new functions related to the control of stomata and trichome number as well as anthocyanin biosynthesis. They started their work by identifying genes responding to the activation of a SPT-GR fusion protein and then characterize process that appear to be affected by SPT. This is a perfectly valid strategy and one that in this case has led to several new discoveries that improve the understanding of this essential transcriptional regulator.

I have only minor suggestions for improvements:

R: Thank you for your kind words, time and help.

Lines 127 – 135: “DNA binding domain” – do you mean “DNA binding motifs”? It would probably important to say that G-boxes are not only bound by bHLH transcription factors but also by bZIPs, etc. From the description it was not clear how putative binding sites were identified. Were they significantly enriched in the dataset or just observed? I recommend clarifying and/or better describing this analysis.

R: Thanks for these comments. We improved the description of the text. Indeed, we meant DNA binding motifs. We rephrased the line G-boxes, which were statistically significant enriched. Furthermore, we improved the M&M a bit. We also noticed that in Fig 1. panel B and C were interchanged; this has now been corrected.

The line now reads: ´The analysis showed the G-box DNA binding motif characteristic for the bHLH transcription factor family to which SPT belongs, in the promoters of……´

Line 160: “SPT is involved in anthocyanin biosynthesis” – this sounds as if SPT played a direct role in anthocyanin biosythesis. It would be probably better to say “SPT is involved in the control of anthocyanin biosynthesis”, or something along those lines.

R: Thanks, we changed the sentence for clarity.

Figure 3: trichome and stomata counts. From panels G-I it appears that Col leaves may be smaller than those of the spl mutant tested. If true, then an increase of stomata number would not be unexpected. In fact, in that case one would conclude that SPT has an effect on leaf size rather than stomata number. Perhaps a calculation of number of stomata per surface area does the trick.

R: Thanks for these comments. We have reanalyzed the data and corrected the Figure 3. There were some errors; but in summary, in the spt mutant, trichome and stomata number is affected. We added also for clarity that all was counted in the same fixed-area, to avoid problems with differences in leaf size.

Spt alleles: is there any reason why different spt alleles have been used for this work? Are the described phenotypes also observed in other spt alleles, i.e. are they allele specific?

R: We did not observe the reduced anthocyanin accumulation in the spt-12 mutant, but only in spt-2. This has been added to the results.

Writing: I feel that the text would benefit from another round of editing – the clarity of the text could be improved in some sections. 

R: Thanks. We read the manuscript various times again and made editing were needed.

Reviewer 2 Report

The manuscript by Bernal-Gallardo et al describes new functions of SPATULA during vegetative development. The authors used an SPT inducible version to identify putative targets in leave tissue. Among them, they identified genes involved in stomata and trichome development. In addition, a subset of genes related to anthocyanin biosynthesis appears to be affected by SPT induction. Mutant plants in the SPT gene and overexpression lines were used to support the transcriptomic results.

The result obtained are very interesting and provide new information on the different roles of SPT during vegetative development. The role of SPT in the control of anthocyanin biosynthesis is strongly supported by the included results. However, concerning the role of the gene in the other two processes, some clarifications and modulation of the conclusion are needed to better fit the obtained results.

Major points:

Q1. The authors indicate in the text that 3 PIF genes and 3 MYC contain G-box binding sites in the promoters and therefore could be putative direct targets of SPT (lines 131-132).

Lines 127-128 indicate that “in silico analysis of the promoters of putative SPT target genes was performed to identify the DNA binding domains enriched in these sequences”. However, searching in the excel file (supplemental data) I could not find any of the PIF genes among the list of “target genes of SPT”. Regarding MYC genes, only MYC3 but neither MYC2 nor MYC4 is present on the list of genes.

Please, explain this discrepancy in the text. Please, also review the related text in the discussion (lines 206-208).

Q2. “Simple Enrichment Analysis” list in the Supplementary file could be improved to make it more informative. Could genes with the same motifs be grouped indicating the type of motif?

Q3. In lines 119 to 122 the authors wrote “Furthermore, by analyzing SPT target genes that do not encode transcription factors, we also identified some genes related to the formation or expressed in stomata, trichomes, and the flavonoid production process, so it was interesting to analyze in more detail these functions of SPT in leaf (Figure 2).” However, figure 2 shows a heatmap of genes including transcription factors (for instance PAP1, TT8, MYB48). Please rephrase the sentence.

Q4. It would be very useful to include a new panel in figure 2 with the anthocyanin biosynthesis pathway highlighting the genes identified in the RNAseq. Some of the genes identified are involved in the early steps of the pathway and other in later stages. Could the latter be indirect targets?

Q5. In figure 3L, the authors analyze the number of trichomes per mm. This quantification reflects trichome density rather than the trichome number. In fact, images of panels G to I in figure 3, do not show changes in total trichome number (10, 11, and 10 respectively). Considering that SPT affects leaf size (reference 21), data need to be reevaluated carefully.

Moreover, in lines 145 to 149 (a single sentence), the authors conclude that SPT positively regulates leaf trichome number disregarding that SPT overexpression has no effect on trichome number. The sentence is contradictory. The results obtained by studying spt mutant and overexpressing SPT lines suggest that SPT is necessary but not sufficient to positively regulate trichome number on its own. Similar results (mild effect of SPT overexpression on stomata number) could be also explained by the lack of an essential interactor.

Please, modify the sentence in the results and discussion related to this subject.

Q6. The authors used two mutant alleles in the experiments, spt-2 and spt-12. Please explain in the text the reason for this choice. What type of mutant are they? Are both null alleles?

Minor points:

M1. Panels A and B from figure 1 have very small lettering and, when enlarged, do not have a good definition. Could be possible to enlarge the size of the panels and/or the quality of the image?

M2. Please include in the graphs the number of samples analyzed for the different experiments (n value).

Author Response

The manuscript by Bernal-Gallardo et al describes new functions of SPATULA during vegetative development. The authors used an SPT inducible version to identify putative targets in leave tissue. Among them, they identified genes involved in stomata and trichome development. In addition, a subset of genes related to anthocyanin biosynthesis appears to be affected by SPT induction. Mutant plants in the SPT gene and overexpression lines were used to support the transcriptomic results.

The result obtained are very interesting and provide new information on the different roles of SPT during vegetative development. The role of SPT in the control of anthocyanin biosynthesis is strongly supported by the included results. However, concerning the role of the gene in the other two processes, some clarifications and modulation of the conclusion are needed to better fit the obtained results.

R: Thank you for your kind words, time and help.

Major points:

Q1. The authors indicate in the text that 3 PIF genes and 3 MYC contain G-box binding sites in the promoters and therefore could be putative direct targets of SPT (lines 131-132).

Lines 127-128 indicate that “in silico analysis of the promoters of putative SPT target genes was performed to identify the DNA binding domains enriched in these sequences”. However, searching in the excel file (supplemental data) I could not find any of the PIF genes among the list of “target genes of SPT”. Regarding MYC genes, only MYC3 but neither MYC2 nor MYC4 is present on the list of genes.

Please, explain this discrepancy in the text. Please, also review the related text in the discussion (lines 206-208).

R: Sorry for the confusion we made, we noticed that in Fig 1. panel B and C were interchanged; this has now been corrected. The transcription factors that putatively bind to the promoters are not in the SPT target gene list, this is correct. These are independent analysis and different results.

Q2. “Simple Enrichment Analysis” list in the Supplementary file could be improved to make it more informative. Could genes with the same motifs be grouped indicating the type of motif?

R: We understand the comment, the results are ordered in the log E-value, so the highest significant DNA binding motif is first mentioned. I think we could leave this in the table.

Q3. In lines 119 to 122 the authors wrote “Furthermore, by analyzing SPT target genes that do not encode transcription factors, we also identified some genes related to the formation or expressed in stomata, trichomes, and the flavonoid production process, so it was interesting to analyze in more detail these functions of SPT in leaf (Figure 2).” However, figure 2 shows a heatmap of genes including transcription factors (for instance PAP1, TT8, MYB48). Please rephrase the sentence.

R: Thank you, we rephrased it by just talking about genes.

Now it reads: ´Furthermore, by analyzing other SPT target genes, we also identified some genes related  in the formation or expressed in stomata, trichomes and in flavonoid biosynthesis,…..´

Q4. It would be very useful to include a new panel in figure 2 with the anthocyanin biosynthesis pathway highlighting the genes identified in the RNAseq. Some of the genes identified are involved in the early steps of the pathway and other in later stages. Could the latter be indirect targets?

R: Thanks for the comment. We thought about adding a panel, but then it would be better to add all three pathways, but it would be make it all very small, so in the end we think it is better not to add the pathways. There are many resources available to quickly check the pathways.

Q5. In figure 3L, the authors analyze the number of trichomes per mm. This quantification reflects trichome density rather than the trichome number. In fact, images of panels G to I in figure 3, do not show changes in total trichome number (10, 11, and 10 respectively). Considering that SPT affects leaf size (reference 21), data need to be reevaluated carefully.

R: We reanalyzed the data and made a new Figure 3. Also we added more clearly that a fixed area was analyzed and not the whole leaf, which indeed has different sizes based on the background. We prefer not to use the word density, because we did not count the number of cells per leaf. The result holds for the mutant that there are less trichomes.

Moreover, in lines 145 to 149 (a single sentence), the authors conclude that SPT positively regulates leaf trichome number disregarding that SPT overexpression has no effect on trichome number. The sentence is contradictory. The results obtained by studying spt mutant and overexpressing SPT lines suggest that SPT is necessary but not sufficient to positively regulate trichome number on its own. Similar results (mild effect of SPT overexpression on stomata number) could be also explained by the lack of an essential interactor.

Please, modify the sentence in the results and discussion related to this subject.

R: Thank you very much for this comment. This is correct, we have added in the Results/Discussion and Conclusion that SPT is not sufficient to regulate trichome number and probably needs an interactor.

Q6. The authors used two mutant alleles in the experiments, spt-2 and spt-12. Please explain in the text the reason for this choice. What type of mutant are they? Are both null alleles?

R: R: We did not observe the reduced anthocyanin accumulation in the spt-12 mutant, but only in spt-2. This has been added to the results. Spt-2 (EMS mutant) is in Ler and spt-12 (T-DNA mutant) is in the Col-0 background. The overexpresses used are in the respective background depending on the mutant allele used.

Minor points:

M1. Panels A and B from figure 1 have very small lettering and, when enlarged, do not have a good definition. Could be possible to enlarge the size of the panels and/or the quality of the image?

R: we increased the resolution, so magnification is possible.

M2. Please include in the graphs the number of samples analyzed for the different experiments (n value).

R: This has been added in the legends and in the M&M.

Reviewer 3 Report

The manuscript entitled “Novel roles of SPATULA in the control of stomata and trichome number, and anthocyanin biosynthesis” submitted by Bernal-Gallardo et al. presents the role of bHLH transcription factor SPATULA in the regulation of stomata and trichome development and anthocyanin biosynthesis.

SPT is crucial to Arabidopsis, especially in leaf development, however, the detailed effect and related mechanism in leaf development are largely unknown. The authors perform transcriptomic analysis in inducible lines in which the downstream genes are affected. They find that genes related to leaf development, stomata formation trichome development, and anthocyanin biosynthesis are highly enriched in induced transcription regulation by SPT. Moreover, authors discover an increased number of stomata and decreased number of trichomes in spt-12 mutants. Furthermore, the authors uncover that sucrose-mediated anthocyanin is also affected in spt-2.

This manuscript integrates the transcriptomic approach and phenotypic analyses and demonstrates the importance of SPT in leaf development. The story is interesting and could attract reader interests in this field. The manuscript is well-written and organized. I have some comments to help the authors to improve the manuscript better.

Minor comments:

1. If authors are willing to do, please add some data from RT-PCR to confirm the expression changes of some important genes in transcriptomic analysis.

2. In Figure 2, some genes are down-regulated such as GCP2 and some genes are up-regulated such as EDA39. Please indicate if target genes are negative or positive regulators in stomata formation.

3. Figure 2, logFC is log10FC or log2FC?

4. In Figure 3G, two yellow dots in the right bottom might be mismarked in the right places.

5. Line 152, in the legend of Figure 3, “(B) Adaxial part of a cotyledon of WT” should be  (B) Abaxial part of a cotyledon of WT”.

6. Figure 4H is a histogram plot, not the box-plot, please correct the statement in line 184. Also, what is the scale bar in the title of the Y-axis?  I think it could be removed.

7. Figure 4H, I just wonder why the authors use a break Y-axis. Each column is not exceeded the bottom edge of the break Y-axis.

Author Response

The manuscript entitled “Novel roles of SPATULA in the control of stomata and trichome number, and anthocyanin biosynthesis” submitted by Bernal-Gallardo et al. presents the role of bHLH transcription factor SPATULA in the regulation of stomata and trichome development and anthocyanin biosynthesis.

SPT is crucial to Arabidopsis, especially in leaf development, however, the detailed effect and related mechanism in leaf development are largely unknown. The authors perform transcriptomic analysis in inducible lines in which the downstream genes are affected. They find that genes related to leaf development, stomata formation trichome development, and anthocyanin biosynthesis are highly enriched in induced transcription regulation by SPT. Moreover, authors discover an increased number of stomata and decreased number of trichomes in spt-12 mutants. Furthermore, the authors uncover that sucrose-mediated anthocyanin is also affected in spt-2.

This manuscript integrates the transcriptomic approach and phenotypic analyses and demonstrates the importance of SPT in leaf development. The story is interesting and could attract reader interests in this field. The manuscript is well-written and organized. I have some comments to help the authors to improve the manuscript better.

R: Thank you for your kind words, time and help.

Minor comments:

  1. If authors are willing to do, please add some data from RT-PCR to confirm the expression changes of some important genes in transcriptomic analysis.

R: Thanks for this suggestion, but at this moment we cannot do these experiments anymore due to time limit.

  1. In Figure 2, some genes are down-regulated such as GCP2 and some genes are up-regulated such as EDA39. Please indicate if target genes are negative or positive regulators in stomata formation.

R: Thank you for this suggestion. In the end we did not add this information, because then we would like to do it for all the genes.

  1. Figure 2, logFC is log10FC or log2FC?

R: It is log2FC, this has been added to the figure.

  1. In Figure 3G, two yellow dots in the right bottom might be mismarked in the right places.

R: Thank you, has been corrected.

  1. Line 152, in the legend of Figure 3, “(B) Adaxial part of a cotyledon of WT” should be  (B) Abaxial part of a cotyledon of WT”.

R: Thank you, has been corrected.

  1. Figure 4H is a histogram plot, not the box-plot, please correct the statement in line 184. Also, what is the scale bar in the title of the Y-axis?  I think it could be removed.

R: Thank you, has been corrected.

  1. Figure 4H, I just wonder why the authors use a break Y-axis. Each column is not exceeded the bottom edge of the break Y-axis.

R: Thank you. The Y-axis break has been moved a bit, but it is to indicate that the intensity scale goes from 0 to 255.